# An Assessment of Mass Flows, Removal and Environmental Emissions of Bisphenols in a Sequencing Batch Reactor Wastewater Treatment Plant

**DOI:** 10.3390/molecules27238634

**Published:** 2022-12-06

**Authors:** Anja Vehar, Ana Kovačič, Nadja Hvala, David Škufca, Marjetka Levstek, Marjetka Stražar, Andreja Žgajnar Gotvajn, Ester Heath

**Affiliations:** 1Department of Environmental Sciences, Jožef Stefan Institute, Jamova Cesta 39, 1000 Ljubljana, Slovenia; 2Jožef Stefan International Postgraduate School, Jamova Cesta 39, 1000 Ljubljana, Slovenia; 3Department of Systems and Control, Jožef Stefan Institute, Jamova Cesta 39, 1000 Ljubljana, Slovenia; 4Wastewater Treatment Plant Domžale-Kamnik d.o.o., Študljanska Cesta 91, 1230 Domžale, Slovenia; 5Department of Chemical Engineering and Technical Safety, Faculty of Chemistry and Chemical Technology, University of Ljubljana, Večna Pot 113, 1000 Ljubljana, Slovenia

**Keywords:** adsorption, biodegradation, contaminants of emerging concern, GC-MS, SBR, sludge, wastewater

## Abstract

This study analyzed 16 bisphenols (BPs) in wastewater and sludge samples collected from different stages at a municipal wastewater treatment plant based on sequencing batch reactor technology. It also describes developing an analytical method for determining BPs in the solid phase of activated sludge based on solid-phase extraction and gas chromatography-mass spectrometry. Obtained concentrations are converted into mass flows, and the biodegradation of BPs and adsorption to primary and secondary sludge are determined. Ten of the sixteen BPs were present in the influent with concentrations up to 434 ng L^−1^ (BPS). Only five BPs with concentrations up to 79 ng L^−1^ (BPA) were determined in the plant effluent, accounting for 8 % of the total BPs determined in the influent. Eleven per cent of the total BPs were adsorbed on primary and secondary sludge. Overall, BPs biodegradation efficiency was 81%. The highest daily emissions via effluent release (1.48 g day^−1^) and sludge disposal (4.63 g day^−1^) were for BPA, while total emissions reached 2 g day^−1^ via effluent and 6 g day^−1^ via sludge disposal. The data show that the concentrations of BPs in sludge are not negligible, and their environmental emissions should be monitored and further studied.

## 1. Introduction

Bisphenols (BPs) are a group of synthetic organic compounds used in the production of epoxy resins and polycarbonate. These polymers are used in various applications, ranging from food contact materials, compact discs, construction materials, thermal paper, dental composites, medical equipment, water pipes, toys and sports equipment [1]. The most widely-known and produced BP is Bisphenol A (4,4′-(propane-2,2-diyl)diphenol, BPA), a known endocrine-disrupting compound. Concerns over its safety have resulted in restrictions on its use in many countries and its gradual replacement by other BPs in specific applications [2]. However, since all BPs share a common structure, i.e., two hydroxyphenyl groups (Appendix A), they also have the potential to be endocrine-disrupting, as is the case for BPS, BPF and BPAF [3].

Bisphenols can enter the environment via different routes, although wastewater treatment plant (WWTP) effluents are considered one of the primary sources [3]. However, their fate during sewage treatment depends on the treatment technology and their physicochemical properties [4]. The end products of wastewater treatment are wastewater effluent and sewage sludge. Treated effluent is commonly discharged to surface water (river or sea) but is sometimes used to irrigate crops in areas that suffer a water deficit [5]. In addition, depending on the WWTP, primary and secondary sludge can be further treated; for example, by aerobic or anaerobic digestion, which are used to stabilize the sludge. In Europe, sewage sludge is either spread on agricultural soils (50%), incinerated (28%), landfilled (18%), or disposed of through other methods (4%) [6].

Although BPs have been investigated in wastewater, a literature review reveals that only a limited number of studies address the presence of BPs in both wastewater (WW) and sludge (Table 1). For instance, only Huang et al. (2021) [7], Sun et al. (2017) [8] and Xue and Kannan (2019) [9] have investigated the fate of BPs during different stages of WW treatment. Huang et al. (2020) [10] report the presence of BPA, BPAF, BPE, BPF, BPS, BPB, BPZ, BPAP, BPP, BPBP, BPC, BPG, BPPH and BPTMC of up to 4537 ng L^−1^ (total) in influent, up to 569 ng L^−1^ in effluent and up to 878 ng g^−1^ in sludge. The most common BPs in influent, effluent and sludge at the highest concentrations were BPA, BPS and BPF. Except for Xue and Kannan (2019) [9], where the removal of BPA, BPF and BPS in two WWTPs was ≤29%, most authors report >87% removal of BPs from WW [10]. According to Hu et al. (2019) [11], the main removal mechanisms are biodegradation and adsorption.

The removal of BPs from sludge during anaerobic digestion remains poorly researched, but limited data suggest low (35%) or negligible removal of BPA, BPS and BPAF [12,13,14]. Interestingly, Abril et al. (2020) [15] found a five-fold increase in BPA concentrations during anaerobic digestion. Overall, the data reveal that BPs are generally more biodegradable under aerobic than anaerobic conditions [11], except for BPAF, which contains strong C-F bonds [4]. Despite these studies, gaps in the knowledge remain, such as the behavior of BPs at different stages of WW treatment and during different sludge treatments. Moreover, among all BPs, only the fate of BPA has been investigated in detail at different points of the technological processes of WW and sludge treatment [12]. In addition, no group has investigated the fate of BPs in WWTP utilizing sequencing batch reactors (SBR), nor the fate of BPs (except BPA) in primary and secondary sludge separately or in anaerobically stabilized sludge.

To address this gap, we studied the fate of BPs in a WWTP utilizing SBR technology and anaerobic sludge digestion. This work involved (1) developing a method for determining 16 BPs in sludge by gas chromatography-mass spectrometry (GC-MS), (2) analyzing their concentrations at different stages of WWTP, (3) determining the adsorption of BPs onto the primary and secondary sludge, (4) calculating their removal from WW and during the anaerobic sludge digestion and (5) evaluating the emissions of BPs into the environment via effluent release and sludge disposal.

**Table 1 molecules-27-08634-t001:** Mean concentrations of BPs in influent, effluent and sludge.

Treatment	Matrix/Removal	Year	Unit	BPA	BPAF	BPE	BPF	BPS	BPB	BPZ	BPAP	BPP	BPBP	BPC	BPG	BPPH	BPTMC	Total	Ref
Primary & secondary treatment, disinfection	Influent	2016	ng·L^−1^	1920.71	1.50	7.13	50.57	85.64	/	/	/	/	/	/	/	/	/	2065.55	[8]
	Effluent		ng·L^−1^	223.71	1.45	8.70	6.69	1.34	/	/	/	/	/	/	/	/	/	241.90	
	SS		ng·g^−1^	445.14	7.14	7.99	70.40	3.40	/	/	/	/	/	/	/	/	/	534.07	
	Removal		%	78.3	−153	−82.5	93.8	98.9	/	/	/	/	/	/	/	/	/	88.3	
Primary & secondary treatment	Influent	2012	ng·L^−1^	60.5	1.1	/	10.4	14.7	2.5	0.6	0.3	7.8	/	/	/	/	/	98.0	[16]
	Effluent		ng·L^−1^	5.2	<LOD	/	0.6	2.4	0.6	<LOD	<LOD	0.8	/	/	/	/	/	9.6	
	PS & SS		ng·g^−1^	5.6	<LOD	/	8.2	185.7	<LOD	<LOD	<LOD	<LOD	/	/	/	/	/	199.0	
	Removal		%	81.6	100	/	96.3	83.1	78.7	100	100	97.6	/	/	/	/	/	90.2	
Primary & secondary treatment, disinfection	Influent	2015	ng·L^−1^	4329	11.7	2.09	71.8	119.6	/	0.71	/	/	0.14	0.32	0.62	0.25	0.78	4537	[10]
	Effluent		ng·L^−1^	548	5.0	2.04	8.67	4.74	/	/	/	/	0.11	0.22	/	0.16	0.25	569	
	Excess sludge		ng·g^−1^	550	5.32	1.24	316	1.07	/	/	/	/	/	0.28	0.43	2.35	1.43	878	
	Removal		%	81	65	2	76	91	/	/	/	/	22	31	/	36	68	87	
Primary & secondary treatment	Influent	2015	ng·L^−1^	71.7	/	/	90.2	29.4	/	/	/	/	/	/	/	/	/	219.5	[9]
	Effluent		ng·L^−1^	39.1	/	/	65.6	25.5	/	/	/	/	/	/	/	/	/	156	
	PS & SS		ng·g^−1^	599.5	/	/	140	11.8	/	/	/	/	/	/	/	/	/	724.5	
	Removal		%	46	/	/	27	13	/	/	/	/	/	/	/	/	/	29	
Primary & secondary treatment	Influent	2015	ng·L^−1^	4121	12.6	3.03	73.3	204	/	0.86	/	/	0.21	0.374	/	<LOD	1.31	4416	[7]
	Effluent		ng·L^−1^	267.8	3.35	<LOD	2.44	0.51	/	0.33	/	/	<LOD	<LOD	/	<LOD	0.68	275	
	PS & SS		ng·g^−1^	275	10.2	<LOD	229	1.50	/		/	/	<LOD	/	/	/	3.11	520	
	Removal		%	94	73	/	97	99	/	62	/	/	/	/	/	/	48	94	

If the authors did not provide the mean values of concentrations or removals, they were recalculated based on the reported data. Removal includes both biodegradation and adsorption to sludge. /: data is not available.

## 2. Materials and Methods

### 2.1. Standards and Reagents

Analytical standards (purity > 98%) of 22BPF, BPAF, 24BPF, 44BPF, BPE, BPA, BPC, BPPH, BPP, BPBP, BPC2, BPZ, BPFL, BPAP and BPS were obtained from Sigma-Aldrich (St. Louis, USA), while BPB was purchased from Dr Ehrenstoffer (Augsburg, Germany). Isotopically labelled ^13^C_12_-BPF, ^13^C_12_-BPS, ^13^C_12_-BPB (CanSyn Chem. Corp., Toronto, Canada) and deuterated BPA-d_16_ (Sigma Aldrich, USA) were used as the internal standards. Acetone (>99.5%, AcO), acetonitrile (≥99.9%, MeCN), ethyl acetate (≥99.5%, EtAc) and methanol (≥99.8%, MeOH) were obtained from J. T. Baker (Deventer, the Netherlands). Concentrated hydrochloric acid (37%, HCl), formic acid (≥98%, FA) and ammonia (≥99.9%, NH_3_) were purchased from Sigma Aldrich (St. Louis, USA). N-Methyl-N-(trimethylsilyl)trifluoroacetamide (>99.0%, MSTFA) and catalyst anhydrous pyridine (Pyridine, 99.8%) were obtained from Sigma Aldrich (Schnelldorf, Switzerland and Steinheim, Germany). Ultrapure water was prepared using the MilliQ-water purification system (Millipore Merck Direct-Q^TM^) to a specific resistance of >18.0 MΩ cm^–1^ at 25 °C. Individual compound stock solutions (≈ 1 mg L^–1^) were prepared in MeOH, while standards and internal standards were prepared by serial dilution from the stock solutions.

### 2.2. Sample Collection

All samples were collected at the Domžale-Kamnik WWTP, Slovenia (149,000 population equivalents), which accepts municipal and industrial WW. The WWTP includes primary treatment (rakes, grease and sand traps, primary settler), secondary treatment (four SBRs) and sludge treatment using anaerobic sludge digestion. The plant accepts external WW and sludge from cesspits and small WWTPs, which enter directly to the primary settler or anaerobic digesters, respectively, and treats hazardous and non-hazardous liquid waste (which are first pre-treated by electrocoagulation) and then enter the mechanical stage. Before disposal, the anaerobically stabilized sludge is centrifuged. The centrate is then treated in a deammonification plant and returned to the mechanical stage. The yearly measurements of the volumetric flows show that external WW from cesspits and small WWTPs, effluents from electrocoagulation and the centrate from mechanical dewatering of anaerobically stabilized sludge represent only 0.25% of the total WWTP influent. For this reason, they were not sampled.

The sampling scheme for BPs monitoring is shown in Figure 1. Twenty-four-hour composite samples (V = 2 L) of WWTP influent (WWTP_inf_), primary settler influent (PSE_inf_), primary settler effluent (PSE_eff_) and WWTP effluent (WWTP_eff_), considering the WWTP hydraulic retention time (24-h), were sampled using automatic samplers in March 2021. Moreover, on the same day, 6-h composite samples (V = 2 L) of primary (PS) and secondary sludge (SS) and a grab sample (V = 2 L) of anaerobically stabilized sludge (AS) were collected manually. Since the sludge retention time of AS in the anaerobic digester is 30 times longer than that of PS and SS, only a grab sample was taken for the preliminary studies. In total, seven samples were collected. The WWTP operational parameters recorded during the sampling period are presented in Appendix A. All of the aqueous samples were stored at −20 °C prior to analysis. Sludge samples were transferred to 50 mL centrifuge tubes and centrifuged at 10,000 RCF for 15 min. The aqueous phase was separated and stored at −20 °C. The remaining sludge was then lyophilized for 72 h at 75 mbar and −55 °C (CHRIST Gamma 1–16 LSCplus). All samples were stored at −20 °C.

### 2.3. Experimental

#### 2.3.1. Extraction of BPs from the Aqueous Phase

Aqueous samples were prepared following the method of Kovačič et al. (2019) [17]. All samples were defrosted overnight and then filtered sequentially through glass fiber (MN GF-2, Machery—Nagel, Dueren, Germany) and cellulose nitrate filters (0.45 μm, Sartorius, Göttingen, Germany) to prevent clogging of the solid-phase extraction (SPE) cartridges. In the case of the aqueous sludge phase, the samples were filtered using MN GF-4 filters. The filtered samples (250 mL) were then spiked with 25 μL of the internal standard mixture ^13^C_12_-BPS, ^13^C_12_-BPF, ^13^C_12_- BPB, BPA-d_16_ (c = 1 μg mL^−1^), and 500 μL of concentrated HCl were added. The samples were loaded onto Oasis Prime HLB cartridges (60 mg, 3 mL; Waters, Massachusetts, USA) at a flow rate of 3 mL min^−1^. After loading, the sorbents were washed with 3 mL of 10% MeOH in water and dried under vacuum (−1.33 kPa) for 45 min. The elution step was performed using 5% FA in EtAc (3 × 0.6 mL). The solvent was evaporated under nitrogen at 40 °C.

#### 2.3.2. Extraction of BPs from the Solid Phase

##### Optimization of SPE (Solid Phase)

The method was optimized for extraction solvent, centrifugation and number of extractions. Prior to clean-up with SPE, two additional clean-up steps using the QuEChERS method (Bond Elut, Fruits and Veg, EN, Agilent Technologies, Santa Clara, CA, USA) and filtration (CHROMAFIL^®^ Xtra PTFE-45/25, Macherey–Nagel, Düren, Germany) were investigated. Full details are in the Appendix A.

Two types of SPE cartridges were assessed for acidified (conc. HCl, pH 2) and non-acidified samples. These included (1) Oasis Prime HLB cartridges (60 mg, 3 mL; Waters, Massachusetts, USA), which are based on water-wettable, hydrophilic-lipophilic-balanced divinylbenzene-N-vinylpyrrolidone copolymer and (2) Affinimip^®^ SPE cartridges designed for extracting bisphenols (100 mg, 6 mL; Affinisep, Petit Couronne, France), which are based on molecularly imprinted polymers. In the case of Oasis Prime HLB cartridges, the tested parameters (washing step and elution solvents) are shown as a schematic in Appendix A.

For the Affinimip^®^ SPE Bisphenols cartridges, two procedures were tested according to the manufacturer’s instructions. In the first procedure (Appendix A), the solvent was evaporated under N_2_ at 40 °C, reconstituted in 1 mL of MeOH/MeCN (1:1, *v/v*), mixed, and 9 mL of water were added. Cartridges were placed on a vacuum manifold (Agilent Technologies) and conditioned with 10 mL of 2% FA in MeOH (*v/v*), 4 mL of MeCN and 4 mL of H_2_O at a flow rate (2 drops s^−1^). After loading the samples at a flow rate of 1 drop s^−1^, the sorbents were washed with 5 mL of water, 3 mL of H_2_O/MeCN (6:4, *v/v*) and 2.5 mL of MeCN (1 drop s^−1^). The elution step was performed using MeOH (2 × 4 mL, 1 drop s^−1^), after which the solvent was removed under N_2_ at 40 °C. Two variations were tested: (1) dissolving the residue in 4 mL of MeOH/MeCN (1:1, *v/v*) and 16 mL of H_2_O, and (2) an additional washing step with 5 mL of H_2_O and 3 mL of H_2_O/MeCN (6:4, *v/v*).

In the second procedure (Appendix A), the solvent was evaporated under N_2_ at 40 °C, reconstituted in 4 mL of MeOH/MeCN (1:1, *v/v*), mixed, and 16 mL of water were added. The cartridges were then conditioned with 3 mL of 2% FA in MeOH (*v/v*), 3 mL of MeCN and 3 mL of H_2_O (2 drops s^−1^). After loading the samples (1 drop s^−1^), the sorbents were washed with 9 mL of water and 6 mL of H_2_O/MeCN (6:4, *v/v*) at 1 drop s^−1^ and dried under vacuum (−1.33 kPa) for 5 min. The elution step was performed using 3 mL of MeOH and 3 mL of MeCN (1 drop s^−1^). The solvent was then removed (N_2_ at 40 °C), and the residue was transferred to a vial (2 mL) using MeOH (3 × 0.5 mL). The solvent was again removed (N_2_ at 40 °C), and the samples were stored at −20°C. Recoveries are given in the Appendix A.

##### Sample Preparation: Solid Phase

Solid phase sludge samples were prepared as follows: a known amount of lyophilized sludge (0.2 g) was transferred to a 10 mL centrifuge tube and spiked with 100 μL of the internal standard mixture ^13^C_12_-BPF, ^13^C_12_-BPS, ^13^C_12_-BPB (c = 50 ng mL^−1^) and 160 μL of BPA-d_16_ (c = 100 ng mL^−1^). An aliquot of MeOH (1 mL) was added, and the contents were homogenized (vortex) for 1 min. The samples were then shaken (oscillating shaker) at 100 rpm overnight, after which 5 mL of MeOH/MeCN (1:1, *v/v*) were added. The samples were then ultrasonicated (VWR MODEL 250D, 180 W) for 15 min, centrifuged at 12,000 RCF for 15 min, and the supernatant was removed. The extraction process was repeated twice, the extracts were combined, diluted with H_2_O (90 mL) and acidified (pH 2) using concentrated HCl (50 µL).

For the extraction step, Oasis Prime HLB cartridges were chosen since they outperformed AFFINIMIP^®^ SPE in terms of recovery (by 4%), repeatability (by 7%), time (2-fold) and cost (3-fold). The samples were loaded onto the cartridges at a flow rate of 3 mL min^−1^, washed with 3 mL of 10% MeOH in H_2_O and dried under vacuum (−1.33 kPa) for 30 min. The elution step was performed using 5% FA in EtAc (3 × 0.6 mL). The extract was then dried under N_2_ at 40 °C.

### 2.4. Instrumental Analysis

The method followed the protocol of Kovačič et al. (2019) [17]. Briefly, before analysis, all samples were derivatized by adding 50 μL of MSTFA and 50 μL of pyridine to the dry residue, followed by heating at 80 °C for 1 h. The samples were then analyzed using a 7890B series gas chromatograph (GC) coupled to a 5977A single quadrupole mass selective detector (Agilent, USA). Chromatographic separation was achieved using a DB-5 MS capillary column (30 m × 0.25 mm × 0.25 µm; Agilent, USA) with helium as the carrier gas (1 mL min^−1^). Samples (1 µL) were injected in splitless mode at 250 °C. The GC oven temperature program was as follows: an initial temperature of 120 °C was ramped at 20 °C min^−1^ to 250 °C and held for 6 min, then ramped to 300 °C at 10 °C min^−1^ and held for 3 min. The total runtime was 22 min. The mass spectrometer was operated in electron impact (EI) mode at 70 eV. The target compounds were identified and quantified using selected ion monitoring (SIM mode). The monitored SIM ions and retention times (RTs) for the derivatized BPs and internal standards are presented in Appendix A. The data were processed using MassHunter software (Agilent Technologies).

### 2.5. Method Validation

Method performance for sludge analysis was assessed regarding recovery, linearity, accuracy, the limit of detection (LOD) and quantification (LOQ), sensitivity, precision (method and instrumental repeatability) and matrix effect. Recovery at low concentrations ranged from 62% (BPBP) to 107% (BPA) and from 81% (BPA) to 131% (BPAP) at high concentrations. The LOD ranged from 0.01 ng g^−1^ (BPPH) to 8.43 ng g^−1^ (BPA), and the LOQ from 0.03 ng g^−1^ (BPPH) to 28.10 ng g^−1^ (BPA). Full details are provided in the Appendix A. Method validation of the aqueous phases was performed previously by Kovačič et al. (2019) [17]. In this case, recovery at low concentrations ranged from 41% (BPAF) to 115% (BPAP) and from 70% (BPPH) to 106% (BPAF) at high concentrations. The LOD ranged from 0.10 ng g^−1^ (BPC) to 5.22 ng g^−1^ (BPA), and LOQ ranged from 0.34 ng g^−1^ (BPC) to 17.39 ng g^−1^ (BPA). Again, full details are provided in Appendix A. All glassware was cleaned according to standard laboratory practice for trace analysis, and procedural blanks were prepared for each experimental setup. Contamination was assessed by evaluating the ratio of the areas between the quantifier ions of the targeted BPs and the internal standards in the procedural blanks, which revealed the presence of BPA and BPS. All sample results were blank corrected, and solvent blanks (EtAc) were analyzed after every tenth sample to evaluate potential carryover.

### 2.6. Calculations of Mass Flows and BPs Removal

The mass flows of BPs in WWTP_inf_, PSE_inf_, PSE_eff_ and WWTP_eff_ were calculated using Equation (1):(1)$${\dot{M}}_{W}=Q_{W}\cdot c_{W}\cdot 10^{-6},$$
where ${\dot{M}}_{W}$ [g day^−1^] is the mass flow, $Q_{W}$ [m^3^ day^−1^] is the volumetric flow and $c_{W}$ [ng L^−1^] is the concentration of BPs in the selected flow in the WWTP. The mass flows of BPs in the primary, secondary and anaerobically stabilized sludge were determined using Equations (2)–(4):(2)$${\dot{M}}_{SP}=Q_{S}\cdot c_{\mathrm{SP}}\cdot \mathrm{TSS}\cdot 10^{-6},$$
(3)$${\dot{M}}_{AP}=Q_{S}\cdot c_{\mathrm{AP}}\cdot (1-\frac{{\mathrm{TSS}}_{\%}}{100\%})\cdot 10^{-6},$$
(4)$${\dot{M}}_{S}={\dot{M}}_{SP}+{\dot{M}}_{AP},$$
where ${\dot{M}}_{SP}$ [g day^−1^] is the mass flow of BPs in the solid phase, ${\dot{M}}_{AP}$ [g day^−1^] is the mass flow of BPs in the aqueous phase, $Q_{S}$ [m^3^ day^−1^] is the volumetric flow, TSS [g L^−1^] and ${\mathrm{TSS}}_{\%}$ [%] are total suspended solids, $c_{\mathrm{SP}}$ [ng g^−1^] is the concentration of BPs in the solid phase, $c_{\mathrm{AP}}$ [ng L^−1^] is the concentration of BPs in the aqueous phase and ${\dot{M}}_{S}$ [g day^−1^] represents the mass flow of BPs in the solid and liquid phases of the sludge. When the concentrations were below the LOQ, a value equal to LOQ/2 [18] was used to calculate the mass flows.

The obtained mass flows allowed the removal of BPs in the SBRs and the anaerobic digester to be estimated by performing a mass balance. The biodegradation of BPs in the SBRs, denoted as ${\dot{M}}_{{\mathrm{RE}}_{\mathrm{SBR}}}$ [g day^−1^], was calculated from the SBR mass balance as follows:(5)$${\dot{M}}_{{\mathrm{PSE}}_{\mathrm{eff}}}={\dot{M}}_{{\mathrm{RE}}_{\mathrm{SBR}}}+{\dot{M}}_{\mathrm{SS}}+{\dot{M}}_{{\mathrm{WWTP}}_{\mathrm{eff}}},$$
where ${\dot{M}}_{{\mathrm{PSE}}_{\mathrm{EFF}}}$ [g day^−1^], ${\dot{M}}_{\mathrm{SS}}$ [g day^−1^] and ${\dot{M}}_{{\mathrm{WWTP}}_{\mathrm{eff}}}$ [g day^−1^] are the mass flows of BPs in the PSE_eff_, secondary sludge and WWTP_eff_, respectively. Similarly, the mass flow of BPs removed during anaerobic stabilization of sludge ${\dot{M}}_{{\mathrm{RE}}_{\mathrm{AS}}}$ [g day^−1^] was calculated from the mass balance:(6)$${\dot{M}}_{\mathrm{PS}}+{\dot{M}}_{\mathrm{SS}}={\dot{M}}_{{\mathrm{RE}}_{\mathrm{AS}}}+{\dot{M}}_{\mathrm{AS}}$$
where ${\dot{M}}_{\mathrm{PS}}$ [g day^−1^] and ${\dot{M}}_{\mathrm{AS}}$ [g day^−1^] are the mass flows of BPs in the primary and anaerobically stabilized sludge, respectively.

The distribution of mass flows between primary sludge, secondary sludge and the WWTP_eff_ was also determined. Since the total mass flow of the primary settler output, i.e., ${\dot{M}}_{{\mathrm{PSE}}_{\mathrm{eff}}}$ and ${\dot{M}}_{\mathrm{PS}}$ was higher than the input mass flow ${\dot{M}}_{{\mathrm{PSE}}_{\inf}}$ [g day^−1^], the sum of output streams was considered a more reliable estimate of the primary settler input mass flow ${\hat{\dot{M}}}_{{\mathrm{PSE}}_{\inf}}$ [g day^−1^]:(7)$${\hat{\dot{M}}}_{{\mathrm{PSE}}_{\inf}}={\dot{M}}_{{\mathrm{PSE}}_{\mathrm{eff}}}+{\dot{M}}_{\mathrm{PS}}.$$
Therefore, the distribution of different streams was determined from the following:(8)$${\hat{\dot{M}}}_{{\mathrm{PSE}}_{\inf}}={\dot{M}}_{{\mathrm{RE}}_{\mathrm{SBR}}}+{\dot{M}}_{\mathrm{PS}}+{\dot{M}}_{\mathrm{SS}}+{\dot{M}}_{{\mathrm{WWTP}}_{\mathrm{eff}}}$$

Finally, the removal of BPs from wastewater ${\mathrm{RE}}_{\mathrm{WW}}$ [%], biodegradation in sequencing batch reactors, ${\mathrm{RE}}_{\mathrm{SBR}}$ [%], removal in the anaerobic digester, ${\mathrm{RE}}_{\mathrm{AS}}$ [%] and adsorption to primary and secondary sludge, ${\mathrm{AD}}_{S}$ [%] were calculated as follows:(9)$${\mathrm{RE}}_{\mathrm{WW}}=\frac{{\hat{\dot{M}}}_{{\mathrm{PSE}}_{\inf}}-{\dot{M}}_{{\mathrm{WWTP}}_{\mathrm{eff}}}}{{\hat{\dot{M}}}_{{\mathrm{PSE}}_{\inf}}}100\%$$
(10)$${\mathrm{RE}}_{\mathrm{SBR}}=\frac{{\dot{M}}_{{\mathrm{RE}}_{\mathrm{SBR}}}}{{\hat{\dot{M}}}_{{\mathrm{PSE}}_{\inf}}}100\%$$
(11)$${\mathrm{RE}}_{\mathrm{AS}}=\frac{{\dot{M}}_{{\mathrm{RE}}_{\mathrm{AS}}}}{{\dot{M}}_{\mathrm{PS}}+{\dot{M}}_{\mathrm{SS}}}100\%$$
(12)$${\mathrm{AD}}_{S}=\frac{{\dot{M}}_{\mathrm{PS}}+{\dot{M}}_{\mathrm{SS}}}{{\hat{\dot{M}}}_{{\mathrm{PSE}}_{\inf}}}100\%$$

## 3. Results and Discussion

### 3.1. Concentrations of BPs in Different Stages of the WW Treatment Process

#### 3.1.1. BPs in the Aqueous Phase

Of the 16 BPs (Figure 2, Appendix A), the most abundant in the WWTP influents/effluents were BPA and BPS and, to a lesser extent, 22BPF, 24BPF, 44BPF and BPE. The highest concentrations were determined for BPS in WWTP_inf_ ≤434 ng L^−1^, PSE_inf_ ≤578 ng L^−1^, PSE_eff_ ≤591 ng L^−1^ and BPA in WWTP_eff_ ≤79 ng L^−1^. Below the LOQ (Appendix A) were BPB, BPC, BPZ, BPFL and BPPH in all water flows, BPBP in the WWTP_inf_, BPC2, BPAF and BPBP in the PSE_inf_, BPC2 in the PSE_eff_ and 22BPF, 44BPF, BPE, BPC2, BPAF and BPP in the WWTP_eff_.

The total concentration of BPs in the effluent from the first part of the mechanical stage (rakes, grease and sand trap) is higher than the BPs levels in WWTP influent (Figure 2). This increase is possible since three potential sources of BPs are introduced to influent prior to the primary settler: (1) external WW (cesspits) delivered by trucks, (2) effluent from a deammonification plant treating centrate from mechanical dewatering of the anaerobically stabilized sludge and (3) pre-treated wastewater deriving from the treatment of hazardous and non-hazardous liquid wastes from the electrocoagulation plant. Moreover, the total amount of BPs is also higher in the effluent from the primary settler. Given that the inflow and outflow should be approximately equivalent and the sample was a 24-h composite sample, a likely explanation for the observed increase is a result of desorption processes from the solid particles, given that anaerobic conditions could occur in the settled sludge at the bottom of the primary settler [19]. Gu et al. (2021) [20] observed a similar trend of increasing concentration through the mechanical stage for BPA.

Concentrations of BPs in WW influent and effluent were much higher than reported by Česen et al. (2018) [2], who analyzed WW from the same WWTP. The obtained BPS concentrations in the influent and effluent are seven times higher compared with literature values (Appendix A), while the obtained BPA concentrations are usually lower than reported. In our study, the levels of BPS in WWTP_inf_, PSE_inf_ and PSE_eff_ are higher than that of BPA. However, this is usually not the case, although Caban and Stepnowski (2020) also observed higher levels of BPS than BPA [21]. There are several possible origins of BPS in WW, but a likely source is from recycled paper used to make toilet paper [22]. However, in the case of the Domžale-Kamnik WWTP, several possible industrial sources have been identified, including wastewater from a paint and lacquer manufacturer, a pharmaceutical factory and a textile cleaning company [2].

#### 3.1.2. BPs in the Aqueous Phase of Sludge

The most abundant BPs in the aqueous sludge phase (Figure 3, Appendix A) were BPA, BPS, 44BPF and 24BPF. The highest concentrations of total BPs were in the anaerobically stabilized sludge, followed by the primary sludge, and the lowest total amount of BPs was in the secondary sludge in both aqueous and solid phases. One reason is that only a single grab sample of anaerobically stabilized sludge was collected, and since its retention time is 30 days, the concentrations cannot be directly related to those in the primary and secondary sludge. Adsorption and desorption of the compounds in the anaerobic digesters, deconjugation of conjugated compounds and possibly the addition of external sludge that goes directly to anaerobic stabilization could also be factors. Moreover, anaerobic digestion of sludge can increase the concentration of trace organic compounds in the anaerobically stabilized sludge due to solids reduction [13].

#### 3.1.3. BPs in the Solid Phase of Sludge

The most abundant BP in the solid sludge phase was BPA and BPS, 22BPF, 24BPF, and 44BPF, to a lesser extent (Figure 4, Appendix A). In other research studies, the levels of BPs are mainly reported in the solid phase of sludge. In the present study, the total concentrations of BPs in the solid phase of primary and secondary sludge are comparable with previous studies (Table 1). Also, the total concentrations of BPs in primary and secondary sludge are lower than those measured in the domestic WWTPs in Korea by Lee et al. (2015) [23]. The authors also suggest that the paper and textile industries are sources of BPA and BPS, while concentrations of BPF are associated with domestic activity. Guerra et al. (2015) [24] reported BPA levels of 1300 ng g^−1^ and 520 ng g^−1^ in stabilized sludge from thermophilic and mesophilic anaerobic digestors, respectively. These values are higher than the 365 ng g^−1^ reported by Abril et al. (2020) [15] in anaerobically digested and dehydrated sludge.

### 3.2. Mass Flows

Mass flows of BPs in different WW and sludge were calculated from their concentrations (Chapter 3.1), volumetric flow, TSS and TSS_%_ (Appendix A). Mass flows were not calculated for the following compounds: (a) BPFL, since its concentrations were below LOQ in all cases; (b) BPB, BPC, BPZ and BPPH, which were detected only in the sludge; and (c) BPC2, since its concentration was <LOQ in the PSE_inf_ and the PSE_eff_. For other compounds <LOQ, the concentrations were substituted with a value equal to LOQ/2 [18]. Therefore, mass flows were calculated for BPS, 22BPF, 24BPF, 44BPF, BPE, BPA, BPAF, BPAP, BPBP and BPP (Figure 5). The data show that the highest bisphenols in the WWTP_inf_ were below 8.15 g day^−1^ (BPS), ≤10.85 g day^−1^ (BPS) in the PSE_inf_, ≤11.09 g day^−1^ (BPS) in the PSE_eff_, ≤1.48 g day^−1^ (BPA) in the WWTP_eff_, ≤1.75 g day^−1^ (BPA) in primary sludge, ≤0.17 g day^−1^ (BPA) in the secondary sludge and ≤4.63 g day^−1^ (BPA) in the anaerobically stabilized sludge. Among all the BPs, the highest mass flows were observed for BPS (≤11.09 g day^−1^) in PSE_eff_ and BPA (≤9.07 g day^−1^) in PSE_eff_, which confirms their high usage [4].

When plotted (Figure 5), the BPs mass flow data indicate a similar trend to BPs concentrations, i.e., increase during the mechanical treatment stage and reaching a maximum in the PSE_eff_. The reason is that the volumetric flows of WW were generalized as one constant volumetric flow. For sludge, the highest mass flows of BPs were present in the anaerobically stabilized sludge, lower in the primary sludge and lowest in the secondary sludge. Possible explanations are single grab sampling of anaerobically stabilized sludge, adsorption and desorption of the compounds in the anaerobic digesters, deconjugation of conjugated compounds, the addition of external sludge and the increase due to solids reduction.

The mass flows of BPA in primary (1.75 g day^−1^), secondary (0.160 g day^−1^) and anaerobically stabilized sludge (4.63 g day^−1^) were high compared to the other BPs. Due to high BPS mass flows in WW, higher BPS mass flows were expected in sludge than were detected. Possible explanations can be a difference in logK_ow_, since BPA has higher logK_ow_ (3.64) and is, therefore, more likely to adsorb to sludge than BPS (logK_ow_ = 1.65). The observation is in contrast to Karthikraj and Kannan (2017) [16], who suggest that BPS has a higher affinity for particulate matter/sludge than BPA, but confirms the results of Xue and Kannan (2019) [9], who observed that BPA has a higher affinity towards particulate matter/sludge than BPS and BPF.

The composition of BPs in selected wastewater flows and sludge (Figure 6) shows that the most abundant BPs in WWTP_inf_, PSE_inf_ and PSE_eff_ are BPS and BPA, while in WWTP_eff,_ BPA is more abundant than BPS. In PS, SS and AS, BPA is more abundant than BPS. In the case of SS and AS, 44BPF is more abundant than BPS, which agrees with Zhu et al., 2019 [25]. The total mass flow distribution between the aqueous and solid phases in primary, secondary and anaerobically stabilized sludge (Appendix A) reveals that most BPs are in the solid phase (85–95%), while only 5–15% are in the aqueous phase of sludge.

### 3.3. Distribution and Removal of BPs from Wastewater

The removal of BPs from wastewater can result from biodegradation, adsorption, chemical degradation or photodegradation, although the latter is unlikely since the sequencing batch reactors are covered at the WWTP Domžale-Kamnik. Biodegradation of BPs (Figure 7) ranged from 45% (22BPF) to 96% (BPE); while between 1% (BPBP) and 45% (22BPF) were adsorbed to the primary sludge, less than 5% (BPAP) adsorbed to secondary sludge, and 2% (24BPF, 44BPF, BPE) to 31% (BPP) remained in the WWTP_eff_. In total, 10% of the BPs were adsorbed to primary sludge, 1% to secondary sludge, 8% remained in the WWTP_eff_, and 81% were biodegraded. Total removal, the sum of biodegradation and adsorption, was 92%, which is consistent with 87–94% removals obtained in China and India (Table 1 and Appendix A) and >96% removal (BPA not included) obtained in Slovenia in a previous study [2]. A comparison of the removals of all BPs with the literature data can be found in Appendix A.

The lowest biodegradation was observed for 22BPF, BPAF, BPAP and BPP. BPAF is a halogenated bisphenol, and its low biodegradation is consistent with compounds with strong C-F bonds being poorly biologically degradable under aerobic conditions [8,17]. BPAP and BPP had a similar biodegradation rate to BPAF, and all had low mass flows. However, the way biodegradation is calculated, considering that LOQ/2 was used when the measured concentration was below LOQ (Chapter 2.6), can affect their calculated distribution, so further investigation is needed to confirm their biodegradation rates. A possible reason for the low biodegradation of 22BPF compared to its other two isomers, 24BPF and 44BPF, could be a consequence of the different positions of the two hydroxyl groups. According to Noszczyńska and Piotrowska-Seget (2018) [1], the degradation pathway of 44BPF described for bacteria *Sphingobium yanoikuyae* FM-2 starts with the formation of bis(4-hydroxyphenyl)methanol after hydroxylation of the 44BPF bridging carbon. Since the two hydroxyl groups of 22BPF are in the second position on the aromatic rings, they are likely to hinder the hydroxylation of the bridging carbon, which may explain why 22BPF is less biodegraded than its isomers.

### 3.4. Removal of BPs in Anaerobic Digesters

Since sludge retention time in the anaerobic digesters (30 days) is much longer than a single treatment cycle of WW (24 h), it is impossible to compare the results of AS mass flow with the PS and SS mass flows directly. Moreover, only a single grab sample of the anaerobically stabilized sludge was taken. Still, measuring the removal of BPs in anaerobic digesters can provide useful information. For instance, the removal of BPAF was among the highest of all BPs, which was expected since halogenated compounds are known to be more biodegradable under anaerobic conditions [26]. The removal of BPS, 22BPF, BPE, BPAF, BPAP and BPP (Table 2) ranged from 11% (BPE) to 82% (BPAP), while an increase in mass flows after anaerobic digestion was observed for 24BPF, 44BPF, BPA and BPBP. The latter is likely due to a single grab sampling of anaerobically stabilized sludge, adsorption and desorption of the compounds in the anaerobic digesters, deconjugation, external sludge addition or, most likely, volatile solids reduction [13]. In other studies, Samaras et al. (2013) [12] reported low removal of BPA (35%), while Phan et al. (2018) [14] reported negligible removal of BPA; in addition to BPA, Choi et al. (2021) [13] reported negligible removal of BPS and BPAF.

Interestingly, Abril et al. (2020) [15] reported a five-fold increase in the levels of BPA during anaerobic digestion, i.e., from 45 ng g^−1^ in primary sludge and 100 ng g^−1^ in secondary sludge to 245 ng g^−1^ in anaerobically stabilized sludge. This finding agrees with a 2.4-fold increase in mass flows of BPA observed in this study, although in our case levels of BPA were much higher. The results also suggest that BPAP and 22BPF are removed more under anaerobic than under aerobic conditions. The reason could be due to the compounds’ structure or unknown anaerobic biodegradation mechanisms.

## 4. Conclusions

This study investigated the occurrence and mass flows of 16 BPs in WWTP based on SBR technology. According to the research goals, this study can be summarized as follows. A method for determining 16 BPs in sludge based on SPE and GC-MS with good average repeatability, recovery and low LOQ values was successfully developed, revealing that BPA and BPS were the most abundant compounds measured in WWTP influent, effluent and sludge and represent 80–90% of all the amounts of BPs in streams. The data also show that the total concentration of BPs increases during the mechanical stage of treatment, with the highest values in the primary settler effluent. In sludge, the highest concentrations of BPs were found in the anaerobically stabilized sludge, followed by primary and secondary sludge. The study also found that significant amounts of BPs remain in the WWTP effluent (8%) and the primary (10%) and secondary sludge (1%), with the majority being biodegraded (81%) in the sequencing batch reactors. Overall, removal was 92%, with the highest daily emissions from the WWTP being 1.48 g day^−1^ and 4.63 g day^−1^ for BPA via effluent and anaerobically stabilized sludge, and the sum of the mass flows of all BPs in the WWTP_eff_ was 2.13 g day^−1^, and in the anaerobically stabilized sludge it was 6.03 g day^−1^. Finally, given the potential toxicity of BPs, these results could prove useful when assessing risk regarding the emissions into the environment and the reuse of wastewater and sludge in agriculture.

## Figures and Tables

**Figure 1 molecules-27-08634-f001:**
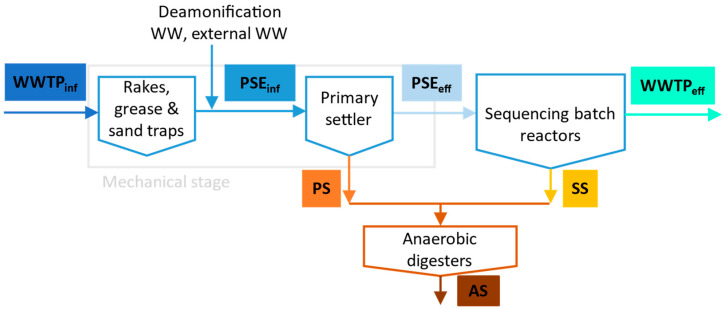
Scheme of the sampling, where sampling points and names of the samples are in colored squares.

**Figure 2 molecules-27-08634-f002:**
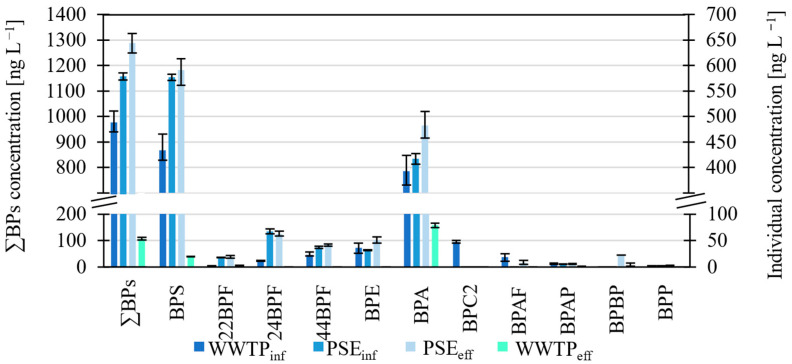
A histogram showing the concentrations of BPs in the WWTPinf, PSEinf, PSEeff and WWTPeff. Compounds are arranged in increasing logKow.

**Figure 3 molecules-27-08634-f003:**
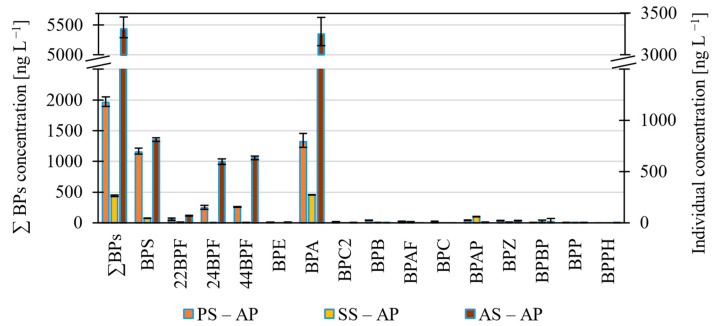
Histogram showing levels of BPs in the aqueous phase of primary (PS–AP), secondary (SS – AP), and anaerobically stabilized sludge (AS– AP).

**Figure 4 molecules-27-08634-f004:**
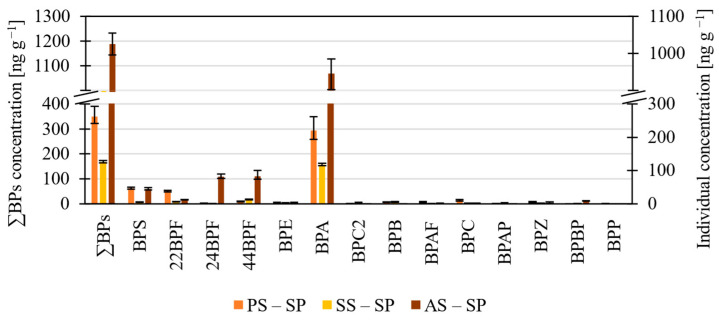
Histogram showing the levels of BPs in the solid phase of primary (PS–SP), secondary (SS–SP) and anaerobically stabilized sludge (AS – SP).

**Figure 5 molecules-27-08634-f005:**
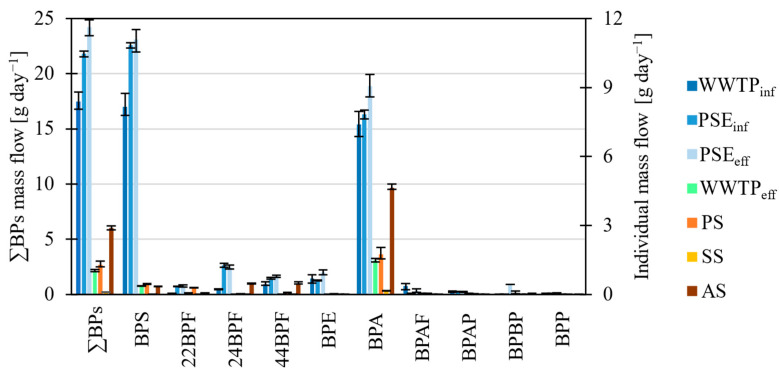
Histogram of mass flows of BPs in the selected wastewater flows and sludge.

**Figure 6 molecules-27-08634-f006:**
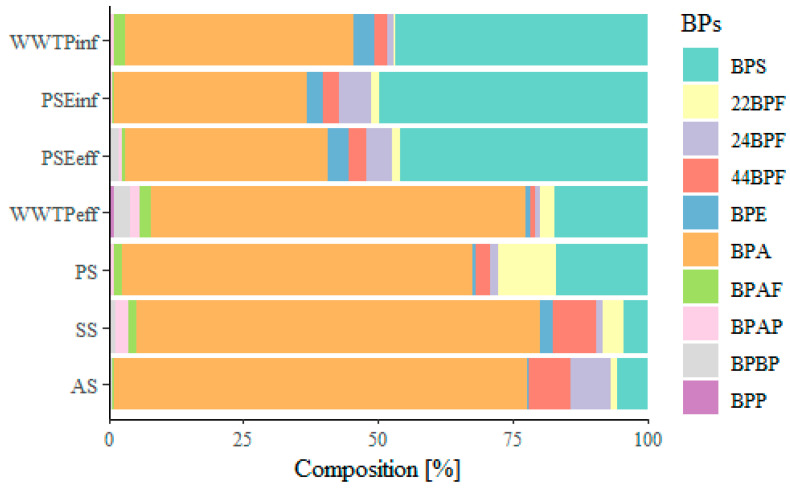
Composition of BPs in the selected wastewater flows and sludge.

**Figure 7 molecules-27-08634-f007:**
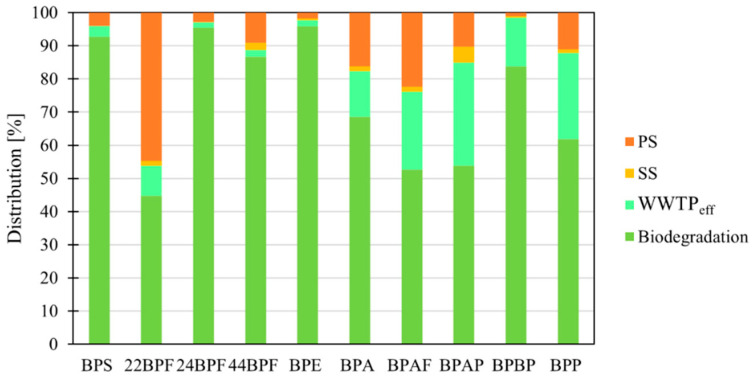
Distribution of BPs in primary and secondary sludge, WWTPeff and their biodegradation.

**Table 2 molecules-27-08634-t002:** Mass flows of BPs in sludge and their removal.

Compound	Mass flow PS + SS [g day^−1^]	Mass flow AS [g day^−1^]	Anaerobic Removal [%]
BPS	0.47	0.33	28
22BPF	0.30	0.07	78
24BPF	0.04	0.45	−1156
44BPF	0.10	0.47	−380
BPE	0.02	0.02	11
BPA	1.91	4.63	−142
BPAF	0.05	0.01	80
BPAP	0.02	0.003	82
BPBP	0.01	0.04	−570
BPP	0.01	0.003	67
Total	2.92	6.03	−107

## Data Availability

All data are available in the manuscript and Appendix A.

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
