# Peer review of "An Assessment of Mass Flows, Removal and Environmental Emissions of Bisphenols in a Sequencing Batch Reactor Wastewater Treatment Plant"

_molecules, 2022, doi:10.3390/molecules27238634_

Round 1

Reviewer 1 Report

In this paper, the authors study the faith of BPs in wastewater treatment plant during wastewater treatment in SBR and anaerobic sludge digestion. Aldo, the topic is interesting, but this paper is bad in presentation. First, the results are presented before the "Materials and Methods" section, which confuses the readers. So, it is hard to understand what was done in the timeline, first, the sampling conditions, because as shown in Fig. 1, the BP values are higher in the PT effluent and influent than in the WW influent, and the authors discuss this as an expected and normal result because there are influents from cesspools and centrifuge effluent returns that mix with the influent WW, and we do not know what the mass flux is. To me, that's a big problem because they had to consider each of those streams and measure the concentrations of the compounds in them and then later calculate and verify the correct mass balance for each compound. They could also use another nutrient to verify the mass balance of the WWTP (e.g. phosphorus) so that the reader understands what is going on in the WWTP and what values were used, and which ones can be trusted. In addition, the authors stated that the goal of the study was stated in five points, and it is not apparent from the conclusion that they met five points of their goals. From all of this, I can conclude that this paper has potential, but it lacks in presenting the experimental design and discussing the results, which is confusing to me in the end. It looks like the authors drafted this paper in a hurry. I can recommend the authors to revise their work thoroughly and after that it could be recommended for publication in Molecules.

Reviewer 2 Report

The manuscript reports Assessing the mass flows and environmental emissions of bi- 2

sphenols during sequencing batch reactors wastewater treatment process. The work looks interesting and the authors have added good technical value to the field and the readers will benefit. However, before publication, it needs to make a revision the research manuscript properly. Therefore, I would like to recommend this manuscript for "Major Revision".

1. The title must be changed to show the paper topic clear.

2. Abstract must be enriched by the important results of the study.

3. I think the keywords in its current form could not satisfied the present research idea, so please revise the keywords to provide complimentary information of the research topics.

4. Authors must modify introduction by stating the following points "Problems, Possible solution, Disadvantages of these solutions, Author's idea, Advantages of authors idea, etc." Introduction in overview form not recommended. Please follow the following references for the modification of the introduction and must cite them.

https://doi.org/10.1016/j.envres.2022.114270

https://doi.org/10.1016/j.jwpe.2022.102696

https://doi.org/10.1016/j.envres.2022.114113

5. The purity and CAS number of each used chemicals must be provided by authors.

6. Authors must compare the results in tabulated form with already published materials so the efficacy of the author's material can be seen.

7. Please provide border for Figures 1-4.

8. The references must be changed based on Molecules Journal.

9. The conclusion is not acceptable in its present form. Therefore, it must be improved by more data.

10. There are several grammatical errors in the manuscript. So, the language must be polished throughout the manuscript before publishing.

11. To make readers more informative of the content of the manuscript, the graphical abstract should be provided by the authors.

Round 2

Reviewer 1 Report

The authors have addressed all the issues raised; I can propose to be accepted for publication in the journal Molecules.

Author Response

REVIEWER 1

Reviewer's comment 1: The authors have addressed all the issues raised; I can propose to be accepted for publication in the journal Molecules.

Reply 1: We would like to thank the Reviewer for reviewing our paper positively for publication.

Reviewer 2 Report

The impact of the manuscript entitled "An assessment of mass flows, removal and environmental emissions of bisphenols in a sequencing batch reactor wastewater treatment plant" does not exhaust and interpretation necessary for the works. The issue was presented in an overly simplified form. The changes introduced by the authors and the answers provided increase the value of the manuscript, but not so much that it could constitute a scientific novelty. I think the authors not paid enough attention to the reviewer’s comments. Also, the paper in its present form not well-qualified and I suppose still there are a couple of errors in this work need to be addressed before considering to be published in Molecules Journal. Therefore, I would like to recommend again "Major Revision".

Author Response

REVIEWER 2

Reviewer’s comment 1: The impact of the manuscript entitled "An assessment of mass flows, removal and environmental emissions of bisphenols in a sequencing batch reactor wastewater treatment plant" does not exhaust and interpretation necessary for the works. The issue was presented in an overly simplified form.

Reply 1: First, we would like to thank the Reviewer for reviewing the paper. We presented the paper in the best possible form we know. We would kindly ask the Reviewer to suggest how to improve the presentation issue and we will be very happy to further improve the manuscript according to reviewers’ exact suggestions.

Reviewer’s comment 2: The changes introduced by the authors and the answers provided increase the value of the manuscript, but not so much that it could constitute a scientific novelty.

Reply 2: We are glad the Reviewer noticed that the changes to the manuscript increased its value. We feel that our manuscript has strong scientific merit (novelty) because of the following reasons:

  • So far, only a small number of bisphenols (predominantly BPA) were analyzed in influent and effluent from conventional wastewater treatment plants, whereas, for the first time, we investigated the 16 bisphenols.
  • Equally, whereas only BPA has been investigated to some degree at different points/stages in the technological processes of WW and sludge treatment, we investigated the fate of 16 BPs at seven different critical points within the WW treatment process and, for the first time in a WWTP using sequencing batch reactors.
  • We also analyzed BPs in primary, secondary and anaerobically stabilized sludge in both aqueous and solid phases, which previously had only been done for BPA.

Novelty is explained clearly in the manuscript:

The removal of BPs from sludge during anaerobic digestion remains poorly researched, but limited data suggest low (35%) or negligible removal of BPA, BPS and BPAF [12–14]. Interestingly, Abril et al. (2020) [15] found a fivefold increase in BPA concentrations during anaerobic digestion. Overall, the data reveal that BPs are generally more biodegradable under aerobic than anaerobic conditions [11], except for BPAF, which contains strong C-F bonds [4]. Despite these studies, gaps in the knowledge remain, such as the behavior of BPs at different stages of WW treatment and during different sludge treatments. Also, among all BPs, only the fate of BPA has been investigated in detail at different points of the technological processes of WW and sludge treatment [12]. In addition, no group has investigated the fate of BPs in WWTP utilizing sequencing batch reactors (SBR), nor the fate of BPs, except BPA, in primary and secondary sludge separately nor in anaerobically stabilized sludge.

To address this gap, we studied the fate of BPs in a WWTP utilizing SBR technology and anaerobic sludge digestion. This work involved (1) developing a method for determining 16 BPs in sludge by gas chromatography-mass spectrometry (GC-MS), (2) analyzing their concentrations at different stages of WWTP, (3) determining the adsorption of BPs onto the primary and secondary sludge, (4) calculating their removal from WW and during the anaerobic sludge digestion, and (5) evaluating the emissions of BPs into the environment via effluent release and sludge disposal.

Reviewer’s comment 3: I think the authors not paid enough attention to the reviewer’s comments.

Reply 3: To the best of our knowledge, we addressed all the reviewer’s comments from the first revision process. The only exception was the inclusion of these suggested references:

  1. Synthesis and characterization of g–C3N4–CoFe2O4–ZnO magnetic nanocomposites for enhancing photocatalytic activity with visible light for degradation of penicillin G antibiotic by Elham Baladi, Fatemeh Davar, Akbar Hojjati-Najafabadi
  2. Magnetic-MXene-based nanocomposites for water and wastewater treatment: A review

by Akbar Hojjati-Najafabadi, Mojtaba Mansoorianfar, Tongxiang Liang, Khashayar Shahin, Yangping Wen, Abbas Bahrami, Ceren Karaman, Najmeh Zare, Hassan Karimi-Maleh, Yasser Vasseghian

  1. Recent progress on adsorption of cadmium ions from water systems using metal-organic frameworks (MOFs) as an efficient class of porous materials by Mojtaba Mansoorianfar, Hafezeh Nabipour, Farshid Pahlevani, Yuewu Zhao, Zahid Hussain, Akbar Hojjati-Najafabadi, Hien Y. Hoang, Renjun Pei

We read the suggested papers with great interest, and they are excellent works. However, we could not find the link between our work and the suggested literature since the papers address the development of novel materials and their use as advanced wastewater treatment technologies. In contrast, our manuscript focuses on the fate of BPs during wastewater treatment with SBR and anaerobic digestion of sludge and their mass flows, removal, and emissions to the environment. However, if we missed reviewers’ suggestions on why and where in the text they should be included, we would gladly revise the manuscript again. Obviously, we will consider them as our work progresses towards improving wastewater treatment for removing emerging contaminants.

Reviewer’s comment 4: Also, the paper in its present form not well-qualified and I suppose still there are a couple of errors in this work need to be addressed before considering to be published in Molecules Journal. Therefore, I would like to recommend again "Major Revision".

Reply 4: Since we would really like to improve the paper in order to be acceptable for publication in Molecules, we kindly ask the Reviewer for more specific comments, so we can directly address the two errors to which he/she is referring. Finally, we again asked two native English speakers, both professionally active in wastewater treatment, to proofread the draft to confirm its correct grammar. The corrections are included in the attached final version of the manuscript.
